# Regulation of GABA_A_ Receptor Subunit Expression in Substance Use Disorders

**DOI:** 10.3390/ijms21124445

**Published:** 2020-06-22

**Authors:** Jeffrey S. Barker, Rochelle M. Hines

**Affiliations:** Department of Psychology, University of Nevada, Las Vegas, Las Vegas, NV 89154, USA; barkej4@unlv.nevada.edu

**Keywords:** GABA, GABA_A_ receptor, expression, transcription, translation, substance use, substance use disorder, alcohol, benzodiazepines, plasticity

## Abstract

The modulation of neuronal cell firing is mediated by the release of the neurotransmitter GABA (γ-aminobuytric acid), which binds to two major families of receptors. The ionotropic GABAA receptors (GABA_A_Rs) are composed of five distinct subunits that vary in expression by brain region and cell type. The action of GABA on GABA_A_Rs is modulated by a variety of clinically and pharmacologically important drugs such as benzodiazepines and alcohol. Exposure to and abuse of these substances disrupts homeostasis and induces plasticity in GABAergic neurotransmission, often via the regulation of receptor expression. Here, we review the regulation of GABA_A_R subunit expression in adaptive and pathological plasticity, with a focus on substance use. We examine the factors influencing the expression of GABA_A_R subunit genes including the regulation of the 5′ and 3′ untranslated regions, variations in DNA methylation, immediate early genes and transcription factors that regulate subunit expression, translational and post-translational modifications, and other forms of receptor regulation beyond expression. Advancing our understanding of the factors regulating GABA_A_R subunit expression during adaptive plasticity, as well as during substance use and withdrawal will provide insight into the role of GABAergic signaling in substance use disorders, and contribute to the development of novel targeted therapies.

## 1. Introduction

Neuronal firing patterns in the brain are powerfully modulated by the inhibitory neurotransmitter γ-aminobutyric acid (GABA). GABA_A_ receptors (GABA_A_Rs) are ligand-gated ion channels that mediate the majority of fast inhibitory neurotransmission in the brain, and GABA_A_R dysfunction is tied to many neurological and psychiatric illnesses, such as anxiety, epilepsy, and the development of substance use disorder. GABA_A_Rs are heteropentamers consisting of five distinct subunits that vary in their expression by brain region, cell type, and subcellular domain, as well as in their function. There are at least 19 receptor subunits, grouped by homologous amino acid sequences into subclasses: *α*1–6, *β*1–3, and *γ*1–3, *δ*, *ε*, *π*, *θ*, and *ρ*1–3 [1]. While the combination and function of receptor subunits varies, most are comprised of two α-, two β-, and one γ- [2,3]. GABA_A_R subunit expression is also temporally regulated, with the expression of GABA_A_R subunit α2, α3, α5, and β3 mRNA predominating during early development, which are superseded by α1, α4, β2 and δ subunit mRNAs in the adult brain [4]. These changes coincide with the switch in the reversal potential for chloride, transitioning GABA from being depolarizing to hyperpolarizing [4,5].

*α* subunits in particular are important determinants of receptor localization and function. GABA_A_Rs composed of *α*1*β*2*γ*2 are the most widely expressed in the adult brain, with α1 being the most highly expressed subunit [2,4,6]. While *α*1, *β*1-3, and *γ*2 immunoreactivities are found throughout the brain, *α*2 subunit containing receptors are enriched in the cerebellum and forebrain including the hippocampus, *α*3 containing receptors are cortically enriched, *α*4 containing receptors are enriched in the striatum and thalamus, and *α*5 containing receptors are enriched in the olfactory bulbs and hippocampus, while α6 immunoreactivity is relatively restricted to cerebellar granule cells and the cochlear nucleus [7,8].

*α* subunit expression also varies by subcellular localization, with α1–3 being predominantly enriched at synaptic sites, while *α*4–6 are often localized extrasynaptically [9]. Synaptic GABA_A_Rs are also targeted to multiple synapse subtypes where interneuron presynaptic compartments contact the dendrites, the soma, or the axon initial segment. The clustering of specific GABA_A_R subtypes to synaptic and extrasynaptic subcellular domains is thought to be regulated by subunit-specific interactions with scaffolding proteins including gephyrin, collybistin, and dystrophin [10,11,12].

Differences in agonist affinity, gating and pharmacological properties have been repeatedly shown by altering subunits of recombinant GABA_A_Rs [13,14,15,16]. Subunit composition also affects the potency of GABA, as *α*2- and *α*3-containing receptors show low potency, *α*1-, *α*4-, and *α*5-containing receptors show intermediate potency, and α6-containing receptors show high potency [17]. GABA_A_Rs are the site of action for many clinically relevant drugs, most of which act as positive allosteric modulators of the receptors and alter the phasic inhibition by prolonging the decay of inhibitory post-synaptic currents (IPSCs), which in turn can prevent neural firing in response to concurrent excitatory stimuli [13,18]. The kinetics of IPSCs at the postsynaptic GABA_A_R is determined by biophysical properties such as the subunit composition and how they cluster at the cell membrane [18]. The specific α subunit is a determining factor in receptor function and setting the kinetics of IPSCs [13,18,19,20].

GABAergic signaling is controlled at the cellular level by changes in neurotransmitter synthesis, vesicular storage, neurotransmitter release and re-uptake, and postsynaptic receptor clustering [1]. GABAergic signaling at the cellular level is often modulated by changes in the expression of GABA_A_Rs. GABA_A_R gene expression is temporally regulated throughout development and the life span, as well as in response to experience, substance use, and as a result of a number of neuropathologies [4,21,22]. The expression of genes encoding the GABA_A_R subunits can be altered at multiple levels including transcription initiation, alternative splicing, mRNA stability, translation, post-translational modifications, intracellular tracking and protein degradation [1] (Figure 1). In the present review, we will discuss what is known about the regulation of GABA_A_R expression across the levels in both normal and pathological states, with a focus on substance use and withdrawal.

## 2. Basal Mechanisms Regulating GABA_A_ Receptor Expression: Transcription, Translation, and Beyond

In order to understand the dynamic regulation of GABA_A_R expression in substance use, it will be important to examine the mechanisms that regulate subunit expression under basal physiological states. The examination of the factors that regulate transcription, translation and beyond reveals numerous levels of regulation, allowing for the precise tuning of GABA_A_R expression. In the following section, we summarize what is known about the transcription factors, epigenetic regulators, and RNA-binding proteins modulating GABA_A_R subunit expression.

The evolution of GABA_A_Rs and their subunits is highly accessible using a phylogenetic tree and the direct examination of gene organization on chromosomes [23]. Many of the subunit genes are organized in *β-α-α-γ* and *β-α-γ* clusters on different chromosomes, thought to have evolved from a single ancestral *β-α-γ* cluster [1,23]. In total, four clusters of genes coding for GABA_A_R subunits have been found in humans, and this organization is thought to be possibly involved in coordinated gene regulation [1,24]. The genes encoding *β*1, *α*2, *α*4, and *γ*1 (*GABRB1*; *GABRA2*; *GABRA4*; *GABRG1*) are all clustered on chromosome 4p14-q12 in humans [25,26]. The *β*1, *α*2, and *α*4 subunits are all enriched in the hippocampus of the adult rat, which may indicate that cluster organization may be necessary for maintaining region-specific expression [1,6]. Chromosome 5q32.1-q35 contains the cluster coding for the most common GABA_A_R subunits *β*2-*α*1-*α*6-*γ*2 [27] (*GABRB2*; *GABRA1*; *GABRA6*; *GABRG2*), again suggesting that the clustering of subunit genes plays a role in the regulation of expression, and that common factors may regulate the expression of multiple subunit genes.

### 2.1. Transcription

A primary control on mRNA levels and gene expression is transcription initiation, which requires the orchestration of multiple transcription factors and other DNA-binding proteins that recognize discrete motifs surrounding the gene, often in untranslated regions (UTRs) [28]. In active chromatin, conserved DNA motifs flanking the 5′ end of genes interact with a diverse set of DNA-binding proteins (Figure 1) that change in response to diverse physiological stimuli. Using genetic approaches in yeast and *Drosophila*, as well as biochemical assays in mammalian cells, large families of sequence-specific activators and accessory factors have been identified that help form the RNA Pol II complex necessary for the initiation of transcription [28]. Despite many advances in our global understanding of transcription control, very little is known about the detailed and elaborate regulation of specific individual genes [28], including the genes encoding GABA_A_R subunits.

In silico analysis and experimentation have been use to examine the promoters of GABA_A_R subunit genes, including the prediction of core elements, possible proximal transcription factors, as well as transcription start sites (Figure 2) [29,30]. The analysis of alignments from multiple species demonstrated a highly conserved 5′ sequence flanking *GABRA1*, including a site for the binding of specificity proteins (Sps), as well as a site for the binding of cAMP response element-binding protein (CREB) [29]. Sps are zinc-finger transcription factors known to regulate the expression of many genes. The neuron-specific transcription factor Sp4 has been shown to regulate the expression of *GABRA1*, as well as excitatory neurotransmitter receptor genes [31]. CREB and inducible cAMP early repressor (ICER) have been experimentally confirmed as transcriptional regulators of the α1 subunit [32]. CREB is a stimulus-induced transcription factor that has been implicated in mechanisms of plasticity [32,33]. CREB has a well-established role in learning and memory [34], and has also been implicated in the response to substance use [35,36].

The *GABRA2* gene has been found to have multiple start sites, three promoter regions, and generate six mRNA isoforms arising from alternative splicing of exons 1A, 1B, and 1C [37]. The exons 1A, 1B, and 1C showed between 61–78% homology to that found in the cloned human DNA [37,38]. Alternative promoters are associated with developmental and tissue-specific gene regulation, which may explain the six *α*2 isoforms, despite no detected difference in the *α*2 protein product [37]. Nucleotide sequence analysis of these 5′-flanking regions of *GABRA2* showed that all three of the alternative promoters are located at guanine–phosphate–cytosine dinucleotide (GpC)-rich regions and that two of the three lacked the typical TATA and CCAAT sequences. In silico comparison of rat and human *GABRA2* confirms a conserved Sp1 site in the 5′ region flanking *GABRA2* [29], which has been experimentally confirmed to be regulated by the neuron-specific transcription factor Sp4 [31].

The *GABRA3* gene is localized to chromosome X, and in a cluster with the *GABRB4* gene [39]. In mice and humans, cDNA showed a similar intron–exon structure between *GABRA3* and other GABA_A_R subunit genes, with a highly unique promoter region containing GA repeats in the core promoter [39]. Multiple repeats vary by species, appear to be random, and generally have minor effects on transcription [39,40]. The *GABRA3* gene appears to have several start sites, yet there is no evidence for more than one promoter [29]. Examination of the *GABRA4* promoter has identified two Sp binding sites that bind Sp3 and Sp4, and are critical for the promoter activity in vivo [41]. The *GABRA4* promoter also contains an early growth response protein (Egr) site that is highly conserved across species [42], and Egr3 has been shown to regulate the change in *α*4 subunit expression following a seizure [42]. Egr family members, especially Egr1, have been shown to be induced in the striatum following substance use [43,44].

Other GABA_A_R subunits have alternative promoters that help to ensure a proper temporal and spatial regulation of gene expression. The *GABRA*5 gene contains at least three different exons in humans that are homologous to those in rats, suggesting an evolutionary importance of this promoter [24]. The sequences of the alternative promoters of *α*5 are not homologous to those of other GABA_A_R subunit promoters, however they do share in common a lack of the canonical TATA and CCAAT boxes, and display cytosine–phosphate–guanine dinucleotide (CpG)-rich sequences [24]. The *GABRA5* promoter contains putative sites for Sp binding, and an activator protein 2 (AP2) motif [29]. GABA_A_Rs containing the α6 subunit show a highly regulated expression restricted to the cerebellum, and the examination of the 5′-UTR points to how this regulation is achieved. A region which enhances *GABRA6* gene expression exclusively in cerebellar granule cells has been found to contain a conserved nuclear factor 1 (NF-1) -binding site [45,46]. Chromatin immunoprecipitation demonstrated that NF-1 binds to the *GABRA6* promoter, and NF1 deletion dramatically reduces the expression of α6 in cerebellar granule cells [45].

In addition to the 5′UTR regulation of transcription, GABA_A_R subunit genes are also regulated by influences on the gene body. For example, the 5′UTR as well as the *GABRB3* gene contain binding sites for methyl CpG-binding protein 2 (MeCP2). MeCP2 is a reader of methylation that recognizes and binds 5-methylcytosine. MeCP2 binding is predominantly thought to lead to the repression of transcription through the slowing of polymerase [47], but has also been suggested to be an activator [48]. Loss of function mutations in the *MECP2* gene lead to the neurodevelopmental disorder Rett syndrome [49,50]. The examination of post mortem tissue from patients with Rett syndrome, and the neurodevelopmental disorder Angelman syndrome, have shown a reduction in *β*3 subunit expression [50,51]. This finding is supported in MeCP2-null mice where the *β*3 subunit is downregulated in the cerebellum. In addition, it has been shown that the *α*1 subunit is downregulated in the frontal cortex, while *α*2 and *α*4 are reduced in the ventrolateral medulla of the MeCP2-null model [51,52]. In an in vitro study, MeCP2 expression activated the expression of the GABA_A_R α1 subunit [53]. The regulation of multiple GABA_A_R subunit expression patterns in tissue from Rett patients as well as models suggest that MeCP2 may be a global transcriptional regulator of GABA_A_R subunit genes [53]. Epigenetic regulation induced via the phosphorylation of MeCP2 has recently been implicated in the gene expression changes that follow exposure to substances [54].

Globally, multiple GABA_A_R *α* subunit genes are flanked by 5′-UTRs that are GpC or CpG-rich and lack the canonical TATA box (Figure 2). *α* subunit genes often have multiple promoters, start sites, and contain consensus sequences for more than one transcription factor, suggesting a high degree of regulation. Genes for *β*- and *γ*-subunits have shown similar features, with some (*β*1, *β*3, *γ*1) lacking the canonical TATA box and being relatively well conserved across species. The elucidation of transcription controls on GABA_A_R subunit expression provides a basis for the generation of mechanistic hypotheses regarding the spatial and temporal regulation of GABA_A_R subunit expression, yet further work is required for the validation of some of the transcription factor consensus sites, and to demonstrate a functional impact on expression. Further investigation may also examine the temporal regulation across development and the relationship to various stimuli. In addition, although many of the above described factors such as CREB, Egr family members, and MeCP2 have been implicated in the response to substance use, studies have not yet examined how these factors may regulate GABA_A_R expression in the maladaptive plasticity that contributes to the development of substance use disorder.

### 2.2. Translation

Following the production of mRNA via transcription, the translation of mRNA into a protein product provides further regulation of gene expression. The translation of mRNAs is a multistep process influenced by a variety of proteins, as well as by features such as the length and structural stability of the 3′-UTR, the amount of mRNA binding sites, and adenylate–uridylate-rich (AU) elements at the 3′-UTR [55,56]. mRNA is assembled into ribonucleoprotein particles consisting of mRNAs and a variety of RNA-binding proteins (RBPs), and the release of mRNA from their cognate RBPs allows for the translation to commence [57,58]. Recent studies have vastly expanded the number of known RBPs by identifying non-canonical RNA-binding domains [59]. The transcriptome-wide identification of RBP mRNA has revealed that most RBP binding occurs in the 3′ UTR of transcripts [60]. In comparison with the rest of the mouse 3′-UTRome, GABA_A_R subunits have a significant increase in the length of their 3′-UTR [55]. An increase in 3′-UTR length is associated with decreased translational activity in HEK cells and human neurons [55,61,62] suggesting a high degree of control over GABA_A_R subunit expression. The length of the 3′-UTR is also extended during development, which indicates an increased translational regulation in mature neurons as compared to developing ones. mRNA for GABA_A_R subunit proteins, scaffold proteins, and transport proteins have been identified as targets for RBPs (Figure 2) [55].

The RBP Pumilio2 (Pum2) is a posttranscriptional regulator that binds an eight-nucleotide consensus sequence in the 3′-UTR of its target mRNA [63]. In a Pum2 knockdown model with characteristics of epilepsy, *GABRA2* mRNA was upregulated two-fold [63]. Prior studies have shown that Pum2 recruits a deadenylase complex that promotes RNA decay, which suggests that mRNA for the *α*2 subunit may become more stable in the absence of Pum2, leading to increased expression [63]. The non-POU domain-containing octamer-binding protein (NONO) is a DNA- and RNA-binding protein from the *Drosophila* behavior/human splicing (DBHS) family of proteins. Mutations in the gene encoding NONO have been linked to intellectual disability in humans [64]. NONO is a member of a neuronal RNA transport complex [57] and is localized to synapses in an activity dependent manner [65]. NONO has been shown to regulate a large number of transcripts, one third of which are synaptic proteins [64]. The effect of NONO on synaptic protein expression is regulated by its RNA-binding domain [64]. A reduction in mRNA and *α*2 subunit protein was detected in the hippocampus of a mouse model with NONO gene disruption [64]. These findings suggest that NONO may be an important promoter of *α*2 subunit mRNA stability and translation. Although no studies have examined Pum2 and NONO in substance use, other RBPs have been implicated in the development of substance use disorders [66], suggesting that the regulation of translation may be an interesting target to explore further.

Fragile X mental retardation protein (FMRP) is a ubiquitously expressed KH-containing RBP that associates with polyribosomes [67]. FMRP has been shown to inhibit the translation of mRNA by directly binding to the ribosome and precludes the binding of tRNA and the translation elongation factors on the ribosome [68]. FMRP has also been suggested to play a role in the trafficking and localization of mRNA to synaptic sites, as well as in synaptic protein synthesis [69]. Some estimates suggest that FMRP targets around 5% of nervous system-enriched genes. A CCG trinucleotide repeat expansion of the 5′-UTR of the fragile X mental retardation 1 (*FMR1*) gene causes the silencing of FRMP, leading to the neurodevelopmental disorder fragile X syndrome [70]. Multiple studies in FMRP knockout mice show a downregulation of *α*1, *β*2 and *δ* subunit expression [71], as well as a reduction in mRNA for the *α*1,3-5, *β*1-3, *γ*1,2 and *δ* subunits with regional specificity [72,73,74]. Together, these results suggest that FMRP may be responsible for regulating the trafficking and/or translation of select subunits of GABA_A_Rs [55,67], although the mechanism by which FMRP regulates subunit expression remains unclear. Recent data have also linked FMRP to substance use, with studies indicating that FMRP acts as a negative regulator of the plasticity induced by exposure to substances [54].

MicroRNAs (miRNAs) are small, non-coding RNAs that bind with 3′UTRs of mRNA and influence gene expression [75,76]. miRNAs exert control over translation via base pairing with complementary sequences within mRNAs, and act to silence gene expression via reducing translation efficiency, destabilizing mRNA, or causing the cleavage of mRNA. In silico analysis predicted binding sites for known miRNAs on the 3′UTR of mRNAs for multiple GABA_A_R subunits [77]. These studies predicted at least six binding sites for miRNAs in *α*1 mRNA, the most abundant number of sites of any of the *α* subunits [77]. In these studies, no binding sites were predicted for *α*2 or 3, and only miRanda predicted sites in *α*4 and 5 [77]. There were multiple miRNA binding sites predicted in mRNA for *β*1-3 subunits [77]. The *γ*2 subunit was predicted to have two sites [77]. In general, the most commonly expressed subunits had the most miRNA binding sites predicted, and those subunits with a more restricted expression had fewer sites, possibly due to shorter 3′UTRs [77]. Experimentally, the mRNA encoding the *α*1 subunit has been shown to be regulated by miR-181a, and increasing levels of miR-181a downregulate *α*1 subunit expression [78]. The expression of miR-186, miR-24, or miR-375 has been shown to downregulate *α*4 expression [79]. miR-203 has been shown to reduce luciferase activity-reporting 3′UTR activity in mRNA encoding the *α*5 subunit [80]. Evidence is accumulating for miRNAs as regulators of GABA_A_R subunit expression, although a number of details remain to be resolved. Evidence has also accumulated for the support of a role of miRNA regulation of gene expression in substance use disorders [81,82,83], which we will discuss in more detail below.

## 3. Experience-Dependent Plasticity Regulates GABA_A_ Receptor Subunit Expression

The ability to continuously adapt to external and internal stimuli is a unique property of the nervous system. The brain is known to adapt to physiological as well as pathological contexts by regulating synaptic connectivity and signaling as a primary response mechanism. Several lines of evidence have demonstrated overlap between the mechanisms regulating plasticity downstream of learning and memory, with the plasticity that is involved in responding to substance use and the development of substance use disorders. 

It is well established that synapses are dynamically regulated by neuronal activity [84,85,86,87]. While many studies have examined experience-dependent excitatory synapse plasticity, less is known about the molecular mechanisms that regulate inhibitory synapses [88,89]. Sensory experience controls multiple steps in the development and refinement of the mammalian brain, many of which stem from the release of glutamate at excitatory synapses and ultimately result in changes in the strength and number of synapses [84,86,88]. The number and strength of inhibitory synapses is in turn influenced by sensory input, as well as the level of excitatory activity, in order to maintain homeostasis [88,90,91,92]. Dynamic regulation of GABAergic inhibition is likely essential to maintaining homeostasis while allowing the network to adapt [93].

CREB is well established to be central in the mechanisms of neuronal plasticity. CREB is a transcription factor that binds to sequences within DNA known as cAMP response elements (CRE), which are found in the 5′UTR of a large number of genes. The production of cAMP or increased intracellular Ca^2+^ activates kinases, that in turn phosphorylate CREB and activate it. Phosphorylated CREB binds the CRE and interacts with the coactivator CREB-binding protein (CBP), which leads to the recruitment of histone acetyltransferases to the promoter. As mentioned above, multiple GABA_A_R subunit genes have consensus sites for CREB binding in their promoter regions. Other downstream targets of CREB activation include immediate early genes (IEGs) that act as a gateway to further genetic regulation. IEG transcription factors such as c-Fos, c-Myc, and c-Jun are widely known as ubiquitous regulators of cell growth and differentiation. Given that the nervous system is continually regulated by stimuli and experience, it is not surprising that a number of IEG transcription factors have been identified and characterized in neurons. Activity-regulated cytoskeleton-associated protein (Arc) and Egr-1, among other IEG transcription factors, have been shown to reflect changes in neuronal activity caused by sensory experience, and their upregulation has been used as a proxy to determine neuronal activity [94,95].

The IEG brain-derived neurotrophic factor (BDNF) is a downstream target of CREB, and is established as a key regulator of GABA_A_R subunit expression. BDNF drives the CREB regulation of *α*1 subunit transcription by activating the Janus kinase (JAK)/signal transducers and activators of transcription (STAT) pathway. The JAK/STAT pathway regulates the expression of BDNF, which ultimately represses *GABRA1* transcription via CREB/ICER binding to the 5′UTR [32]. BDNF also modulates expression of the *α*4 subunit through activating the protein kinase c (PKC)/mitogen-activated protein kinase (MAPK) pathway and modulating levels of Egr3. In addition to *α*1 and 4 subunits, BDNF is also implicated in modulating the expression of *α*2, *β*2, *β*3, and *γ*2 subunits [96]. BDNF is a well-known contributor to neurodevelopment, as well as to plasticity induced by learning and memory [97], and substance use [98,99].

More recently, the IEG transcription factor neuronal PAS domain protein 4 (NPAS4) has gained attention as an important regulator of inhibitory synaptic connectivity downstream of neuronal activity. NPAS4 is expressed exclusively in response to membrane depolarization, making it explicitly experience-dependent [88,89]. In hippocampal pyramidal neurons, NPAS4 regulates inhibitory synapse number through biasing inhibitory synaptic input to the cell body, away from the apical dendrites. This effect on subcellular targeting has been shown to be mediated by the NPAS4 target gene BDNF [100]. Sensory enrichment increases NPAS4 levels in CA1 of the mouse hippocampus, and recruits synapses made by inhibitory cholecystokinin (CCK) basket cells, but not parvalbumin basket cells [100]. Increasing the number of contacts from CCK basket cells presumably results in the increased expression of GABA_A_Rs, that are enriched at these sites on the pyramidal cell soma, but the mechanisms of this subunit regulation remain to be elucidated.

## 4. Substance Use Regulates GABA_A_ Receptor Subunit Expression

Persistent substance use induces pathological plasticity in the brain’s reward system and converts the normal reinforcement of homeostatic behaviors into compulsive drug seeking. Broadly, substance use disorder is defined as an inability to control the use of a substance or drug, either illegal or legal, despite harmful consequences. Substance use disorders are diagnosed in the fifth edition of the Diagnostic and Statistical Manual of Mental Disorders (DSM-5) by 12 criteria that include hazardous use, neglecting major roles to use, tolerance, withdrawal, and cravings, amongst others [101]. Meeting more than two criterion constitutes a substance use disorder, with the severity increasing as more criteria are met. Tolerance refers to the adaptive response of the body and the nervous system to repeated substance use, where reduced responsiveness is seen for successive equal doses. Withdrawal occurs upon the abrupt removal of the substance, and involves key motivational elements such as chronic irritability, emotional pain, malaise, dysphoria, stress, and loss of motivation for natural rewards [102]. Repeated use of commonly abused substances causes positive reinforcement through dopamine (DA) signaling in the mesolimbic pathway, primarily from the ventral tegmental area (VTA) to the medial shell of the nucleus accumbens (NAc) [103,104]. This pathway is implicated in the reinforcing effects of abused substances, as well as natural reinforcers such as food [103,105]. Repeated substance use potentiates excitatory afferents onto DA neurons in the VTA, and can cause synaptic plasticity in the NAc [106]. Circuit adaptations also induce the aversive response to the abrupt termination of long-term drug use through stress mediators such as corticotropin-releasing factor and noradrenaline in the extended amygdala, as well as adaptations by the NAc [103,105]. Transcriptional and post-transcriptional gene regulation are thought to play a large role in the changes in brain function related to developing and maintaining substance use disorders [55,107].

Despite a lack of direct action on DAergic signaling, many of the drugs that positively modulate GABA_A_Rs are prone to abuse. The action of GABA on GABA_A_Rs is well known to be modulated by a variety of pharmacologically and clinically important drugs including benzodiazepines (BZs), barbiturates, steroids, and anesthetics [2]. Generally, these drugs act as positive allosteric modulators (PAMs) at unique sites on the GABA_A_R, resulting in alterations to the effects of GABA binding [108]. BZs typically act as PAMs of GABA_A_Rs by binding to a specific BZ site in the extracellular domain between *α* and *γ* subunits [108]. GABA_A_Rs are also the targets of general anesthetics such as propofol, etomidate, and halothane, with different anesthetics possessing distinct subunit selectivity and binding sites [108,109]. Alcohol is also thought to act upon GABA_A_Rs, stemming from the sensitivity of the GABA inhibitory Cl^-^ current to alcohol use at low doses (≤30 mm), although some controversy exists about the precise mechanism of the modulation [108,110,111,112]. Many studies have examined the action of GABA_A_R PAMs, yet many questions remain about how the positive modulation of GABAergic signaling leads to the activation of the DAergic reward pathway, and the precise role for GABAergic signaling in the development and maintenance of substance use disorders.

### 4.1. Benzodiazepine Use

GABA_A_Rs are the primary binding sites for BZs, which are commonly used to treat disorders such as anxiety, insomnia, and epilepsy [113]. GABA_A_Rs composed of *α* (1,2,3, or 5), along with *β* and *γ* subunits, are considered BZ-sensitive receptors [114]. BZs modulate the Cl^−^ conductance of GABA_A_Rs by increasing the frequency at which the ion channel opens [115]. BZs are highly prescribed but are associated with a number of side effects, and also display a rapid development of tolerance, produce dependence and withdrawal symptoms, and are commonly abused substances. The development of tolerance, dependence, and withdrawal have been associated with changes in GABA_A_R expression, binding, and function (summarized in Table 1).

With BZs, the tolerance to anxiolytic effects has been shown 48 hours after even a single dose of lorazepam in mice [127]. The mechanisms underlying the development of tolerance are still not completely known, but altered GABA_A_R subunit gene expression downstream of repeated BZ use has been demonstrated. GABA_A_R subunit expression measured by quantitative real-time PCR during the chronic administration of BZs in vivo demonstrated a 50% decrease in the levels of cortical *α*1 subunit mRNA after 14 or 28 days of administration of the full PAM lorazepam [116]. In contrast, fewer lorazepam administration days (1, 2, 4, 7, and 10) did not modify *α*1 subunit mRNA [116]. The *γ*2 subunit mRNA shows a similar time-course and level of reduction in response to chronic lorazepam treatment [116].

Epigenetic mechanisms may be involved in the downregulation of GABA_A_R subtypes that lead to the development of tolerance, particularly to full PAMs that have been well established to elicit these changes, through DNA methylation, histone acetylation and methylation, chromatin modifications and other lesser understood mechanisms [117]. Epigenetic regulation of gene expression is implicated in psychiatric disorders, including substance use disorders, as well as in nervous system adaptations to substance use such as tolerance, dependence, and withdrawal [128]. The partial PAM imidazenil does not produce tolerance for the anti-convulsant effects of BZs with chronic use and does not lead to a decreased expression of mRNA for the *α*1 subunit seen with full PAMs [117]. The lack of effect on *α*1 subunit expression with partial PAMs may be due to differences in histone acetylation at the promoter region upstream of *GABRA1* [117]. The full PAM diazepam showed decreased acetylated histone H3 at the promoter, and increased MeCP2 occupancy in the promoter region [117]. The decrease in H3 acetylation was due to an increase in the expression of histone deacetylase (HDAC) enzymes HDAC1 and HDAC2 [117]. Class I HDACs are often found in a complex together with MeCP2 that acts as a transcriptional repressor at the promoter of a gene, suggesting a possible mechanism of *α*1 subunit downregulation when given a chronic administration of full PAMs at the BZ site [117,129,130]. The assessment of total acetylated histone H3 protein verified that the chronic use of diazepam was not reducing the overall H3 protein levels in the cortex, and the reduction was specific to that bound to the promoter for *GABRA1* [117].

Beyond subunit expression, multiple studies have shown evidence for the uncoupling of GABA and BZ binding sites wherein allosteric enhancement is reduced without changes in binding affinity [113,131,132,133]. When the full PAM diazepam or partial PAM Ro 16 6028 are administered, rat neurons in vitro showed a 50% reduction in receptor sensitivity when administered GABA [134]. Early experiments used embryonic chick neuronal cultures to show a similar rapid development of tolerance to the BZ flurazepam via uncoupling of about 34% in 18h [131]. An experiment in which rat cultured cortical neurons were administered diazepam for 48 h and measured for potentiation via a binding assay showed the uncoupling of the sites [135]. Furthermore, uncoupling was prevented by use of the BZ site antagonist flumazenil, or picrotoxin, a GABA_A_R channel blocker, or nifedipine, an inhibitor of L-type voltage gated Ca^2+^ channels, all pointing towards a mechanism that involves the binding of diazepam to a specific site and the subsequent activation of a GABA_A_R.

The surface level of postsynaptic GABA_A_Rs is related to the strength of synaptic inhibition and is modulated by regulated trafficking steps to and from the plasma membrane. Receptor removal, degradation, insertion and diffusion have all been shown to be dynamically regulated [136]. Jacob et al. looked at GABA_A_R surface levels after treatment with the BZ flurazepam, and demonstrated a dramatic decrease in *α*2 subunit-containing GABA_A_R surface expression along with total levels of the subunit [113]. BZ treatment did not alter the insertion and endocytosis rates of *α*2-containing GABA_A_Rs, but did promote degradation, which was reversed by blocking lysosomal degradation [113]. This loss in total GABA_A_R levels may begin the series of adaptations that contribute to tolerance, as degradation occurs over the first 24 h of treatment.

### 4.2. Alcohol Use

Alcohol use produces anxiolytic, anticonvulsant, sedative-hypnotic, cognitive-impairing, and motor coordination-impairing properties, similar to other drugs that act on the GABAergic system. The mechanisms behind the effects of alcohol include direct and indirect effects on GABA_A_Rs, as well as modulation of GABA release and the synthesis, and availability of endogenous neuroactive steroids [120], as well as actions on other neurotransmitter systems. Research on the exposure to alcohol has shown that there has to be epigenetic mechanisms involved in alcohol use and the development of alcohol use disorder [137,138]. In search of a reliable bio-marker for chronic alcohol use, an epigenome-wide association study was performed looking at the differing methylation of CpG sites in relation to alcohol consumption levels across 13 different cohorts [139]. They found 144 CpGs to be associated with alcohol consumption, and that epigenetic changes in GABA_A_R and GABA_B_R subunit genes were significantly associated with the expression level of a number of genes that are involved in immune function [139]. Of the CpGs most significantly tied to alcohol use, cg04781796 is located on a CpG island intronic to the GABA_A_R *δ* subunit (*GABRD*), and cg09577455 is located on a CpG island intronic to the GABA_B_R subunit 1 (*GABBR1*), and both are implicated in immune function [139]. Similar to BZs, the modulation of histone acetylation caused by a decrease in HDAC activity is seen with alcohol use, and is thought to take part in the anxiolytic effects of alcohol consumption [119]. The CREB target-genes BDNF, Arc, and neuropeptide Y are all increased in the amygdala with acute alcohol exposure [36], and may initiate cascades that result in the regulated expression of GABA_A_R subunits.

Long-term alcohol administration has been shown to cause differential changes in the expression of GABA_A_R subunit mRNA and protein levels in various brain regions (summarized in Table 1) [120]. mRNA for the *α*2 subunit is down-regulated in the central amygdala of human alcoholics [140]. RT-PCR studies in humans have shown increased *α*1 expression in the frontal cortex of alcoholics, as well as total *α* subunit concentration compared to controls [120,121,141]. In contrast, rodent studies have shown a decrease in *α*1 expression and an increase in *α*4 expression with chronic alcohol exposure [120,122,126]. In a chronic intermittent ethanol (CIE) rat model, as well under acute alcohol doses, *α*2-containing GABA_A_Rs are upregulated in the hippocampus, basolateral amygdala, and NAc, all of which are implicated in regulating addiction [126]. Chronic exposure to alcohol in cynomolgus macaques reduces mRNA for *α*2 and *α*3 subunits, and increases *α*1 subunit mRNA in the basolateral amygdala [142]. Chronic alcohol exposure also affects GABA_A_R subunit mRNA expression in different cortical areas of cynomolgus macaques [143]. mRNA for *α*2, *α*4, *β*1, *β*3, *γ*1 and *γ*3 subunits are significantly reduced in the orbitofrontal cortex, while *β*1, *β*2, *γ*1 and *δ* subunit mRNAs are reduced in the dorsolateral prefrontal cortex [143].

Rodent mutant models lacking specific GABA_A_R subunits show changes in physiological and behavioral responses to alcohol consumption [144,145,146,147,148,149]. *α*1-null mutant mice show a decrease in alcohol consumption and an increased aversion to alcohol [144]. *α*1 and *β*2 null mutants experience changes in the sedative effects of alcohol and display shorter periods of loss of righting reflex following consumption [145]. *α*1 and *β*2 knockout mice also show increased locomotor stimulant effects from alcohol exposure [144,146,147]. A knockdown for *α*2 also displays a reduction in binge-like drinking [149]. Using viral-mediated mRNA interference to reduce *α*4 subunit expression in the NAc shell has been shown to reduce alcohol consumption at low-to-moderate doses, indicating that the *α*4 subunit may be involved in developing a preference for alcohol [150]. Mice lacking the *δ* subunit exhibit less of a preference for voluntary alcohol consumption [148]. *δ* subunit knockout mice also have reduced hyperexcitability from withdrawal after chronic alcohol exposure [148]. Studies in animal models and human alcohol use disorder populations, combined with genetic association in families with multiple alcoholics, identify the regulation of GABA_A_R *α* subunit expression as a contributor to alcohol use disorder [151,152].

miRNAs and other non-coding RNAs are emerging as targets in substance use disorders stemming from their role in experience-dependent synaptic plasticity [76,137]. miRNA analysis in frontal cortex tissue from human subjects with alcohol use disorder revealed the upregulation of about 35 miRNAs compared to the controls [153]. Target prediction and classification suggested that mediators of synaptic plasticity are major targets of the detected miRNA alterations [153]. Similarly, in animal models of chronic alcohol exposure, alterations in the expression of over 100 mRNAs and ~30 miRNAs were detected with some regional specificity [154]. Gene ontology and pathway analysis suggested that major targets were involved in synaptic plasticity among other cell signaling pathways [154]. Although GABA_A_R subunits have not been identified as major targets of miRNA alterations in chronic alcohol exposure, related pathways and known regulators such as BDNF have been identified across multiple studies.

In addition to the regulation of expression levels, chronic alcohol exposure has shown a reduction in *α*1 subunit-containing GABA_A_R surface expression in both the cerebral cortex and the hippocampus [155]. An experiment in cultured rat neurons from the cerebral cortex showed a dose-dependent (50 mm) reduction in surface *α*1 subunit-containing GABA_A_Rs after 4h of acute ethanol exposure as measured by biotinylation and Western blotting [156]. The 4 h exposure also reduced the GABAergic response to the subunit selective PAM zolpidem, further showing a functional impact of reduced surface expression [156]. The mechanism behind the internalization of GABA_A_Rs has been linked to alterations in the expression and localization of protein kinase C (PKC) [156]. Phosphorylation is a well-known regulator of GABA_A_R function under various physiological and pathological conditions [157,158,159], including alcohol use disorder [160]. GABA_A_Rs have a consensus site for protein kinases PKA and PKC, located on the *β* subunit whose phosphorylation can lead to changes in GABA binding, conductance, and possibly internalization [161,162]. A PKC activator mimics the effects of ethanol exposure on GABAergic signaling, and produces a similar reduction in surface level receptors [156]. The internalization of *α*1 subunit-containing GABA_A_Rs was prevented by combining the PKC inhibitor calphostin C with ethanol exposure [156]. Immunoprecipitation in conjunction with ethanol administration showed increased association between *α*1 subunit and PKC*γ*, but not PKC*β* [156]. In addition, mice lacking the isoform PKCϵ show an increased behavioral response to ethanol through reduced *γ*2 subunit phosphorylation [163].

### 4.3. Use of Other Highly Abused Substances

While stimulant drugs do not directly bind to GABA_A_Rs, the use of stimulants including nicotine [164,165], cocaine [166,167,168], and amphetamines [169] has been associated with changes in GABA_A_R subunit expression. With similarity to families with alcohol use, several single nucleotide polymorphisms in *GABRA4* and *GABRA2* have been linked with an increased likelihood for nicotine dependence [164]. The *α*2 subunit is thought to lead to nicotine dependence by the activation of Toll-like receptor 4 in VTA neurons, leading to the activation of CREB and the upregulation of corticotropin-releasing factor and tyrosine hydroxylase, which play a role in the sensitization to and reinforcing effects of nicotine [165]. In samples from the hippocampus of human cocaine addicts, *GABRG2* is down-regulated along with the gene encoding the GABA_A_R-associated protein gephyrin (*GPHN*) [168]. *GABRG2* and *GPHN* are both up-regulated in the hippocampus of alcohol-preferring rats, which also show increased nicotine self-administration and cocaine-seeking behaviors [168]. Other studies have shown that the deletion of the *GABRA2* gene abolishes the ability of cocaine to facilitate conditioned reinforcement [166]. In animal models, *α*2 subunit levels are decreased in the NAc shell with sensitization to cocaine, but not the NAc core [167]. Sensitization to methamphetamine in rats leads to decreased *α*2 expression in the NAc core and shell, and increased expression in the caudate nucleus [170]. Methamphetamine-sensitized rats also show the upregulation of mRNA for α3 and β1 subunits in the prefrontal cortex [169]. 

In addition to stimulant use, there are indications that opiate exposure may also induce alterations in GABA_A_R subunit expression [171]. A microarray study of gene families suggests a mix of the induction and repression of specific subunits in the NAc during a 14-day morphine exposure paradigm [172]. Overall, *α* and *β* subunit expression was somewhat repressed, while *γ*, *δ*, and ε subunit expression was enhanced, particularly around 8 days of morphine exposure [172].

### 4.4. Withdrawal

As we have described above, GABA_A_R expression and function have been shown to change in response to substance use, and contribute to tolerance and the development of substance use disorder. Withdrawal from an abused substance has also been shown to modify the expression and function of GABA_A_R subunits [123]. To observe the effects of withdrawal, hippocampal neurons in culture were given 5 days of continuous exposure to ethanol followed by a non-ethanol medium for 3–24 h, and mRNA levels, neuronal morphology, and the functional and pharmacological responses of GABA_A_Rs were examined. Ethanol treatment alone induced decreases in mRNAs for *α*1, *α*3, *α*4, and *α*5, as well as two variants of the *γ*2 subunit, while not causing a change in *α*2 mRNA levels [123]. During withdrawal, however, *α*2, *α*3, and *α*4 all significantly increased, peaking at around 3 h after alcohol removal, while *α*1 and *γ*2 subunits had returned to baseline by 9–12 h after ethanol removal [123]. To observe the pharmacological responses during withdrawal, cultured neurons were incubated in media containing compounds shown to reduce withdrawal symptoms in human subjects with alcohol use disorder and alcohol-dependent laboratory animals, including diazepam, gamma-hydroxybutyrate, or the GABA_B_R agonist baclofen. Both diazepam and gamma hydroxybutyrate mitigated the withdrawal-induced increase in *α*2, *α*3, and *α*4 subunits [123]. The BZ diazepam, a full PAM, also shows increased *α*4 mRNA and total protein upon 6 h of withdrawal, coupled with decreases in *α*1 and *γ*2 subunits [118]. Interestingly, the partial PAM imidazenil that does not cause robust changes in subunit expression during use, still causes a comparable increase in *α*4 expression and decreased *α*1 and *γ*2 expressions during withdrawal [118]. In addition to withdrawal from alcohol and BZs, alterations in the expression of GABA_A_R subunits have been noted in opiate withdrawal. In morphine-tolerant rats, withdrawal induced an upregulation of ε subunit mRNA in the locus coeruleus [173], while a microarray study suggested a relatively broad downregulation of GABA_A_R subunit genes in the NAc [172]. Between 4 and 18 days of abstinence, there was a repression of the *α*, *β*, *γ*, *δ*, and ε subunits assayed [172]. These results suggest that changes in GABA_A_R subunit expression may be a common feature of withdrawal plasticity.

Withdrawal from alcohol has also been shown to lead to the decreased phosphorylation of CREB, decreased histone H3 and H4 acetylation, and decreases in BDNF, Arc, and neuropeptide Y in the amygdala, an area of the brain implicated in the anxiety induced by withdrawal [36]. These changes are the opposite of what occurs during acute exposure, and are thought to be caused by increased HDAC activity upon withdrawal [36]. Inhibiting HDAC activity in rats reduces alcohol withdrawal-induced hyperalgesia, suggesting that epigenetic modifications stemming from withdrawal can also alter pain processing [174]. Protracted abstinence from chronic alcohol exposure in rats has also revealed alterations in miRNA levels in the frontal cortex [175]. In this study, over 40 rat miRNAs were found to be altered in the frontal cortex in the protracted abstinence phase, along with alteration in approximately 165 mRNAs [175]. Using miRNA–mRNA expression pairing revealed 33 miRNAs putatively targeting 88 mRNAs, many of which were involved in the regulation of synaptic signaling, including BDNF [175].

Taken together, these data clearly support an alteration in GABA_A_R expression as a downstream effect of substance use, as well as in response to withdrawal from an abused substance. In addition, changes in GABA_A_R subunit expression are not only seen with exposure to substances that directly act on the GABAergic system, but also with exposure to other highly abused substances like stimulants and opiates. The changes in GABA_A_R subunit expression vary by subunit, substance under study, experimental paradigm, time point, and brain region, adding much complexity to the picture. Changes in specific brain regions such as the NAc and VTA are associated with drug reward, while changes in cortical areas such as the prefrontal cortex, may be related to drug seeking and choice behaviors. The mechanisms that lead up to the changes in GABA_A_R expression are diverse, yet some common factors emerge—there is a compelling overlap with the mechanisms and factors that underly plasticity in response to learning and memory (CREB and BDNF), as well as with factors that relate to neurodevelopmental plasticity (MeCP2 and FMRP). Regardless of the complexity, changes in GABA_A_R expression appear to be a consistent hallmark of the maladaptive plasticity that occurs with repeated exposure to substances of abuse as well as withdrawal, warranting further detailed study. Future studies may endeavor to link changes in specific subunits, across multiple brain areas, in association with specific timepoints following exposure. It will also be important to examine specific facets of substance use disorder including reward and reinforcement, motivational and hedonic influences, as well as contributions from stress and anxiety, which are known contributors to the complex picture of substance use.

## 5. Conclusions

GABA_A_Rs are highly specialized receptors comprised of a variety of subunits that vary in expression based on their location and function [2,3]. The expression of specific subunits occurs in an adaptive and plastic manner, responding to stimuli ranging from sensory experience to substance use and neurodegeneration. Different classes of sunstances such as BZs and alcohol act upon and modulate the expression of GABA_A_Rs by altering the expression with regional and subunit selectivity, as well as altering receptor surface expression [113,116,117,120,122,126,155]. Transcriptional regulation of mRNA is the most prevalent means of changes in expression, but the factors controlling the transcription of individual GABA_A_R subunit genes are not completely known. Due to their clustered position on their respective chromosomes, GABA_A_R subunits are thought to have evolved from a single gene, and most GABA_A_R subunit genes are flanked with 5′-UTRs that are CpG-rich and lack the canonical TATA box [1,23,24]. CREB and BDNF are thought to be important and interrelated regulators of the expression of multiple GABA_A_R subunits, via multiple signaling pathways [96,176]. Both of these proteins are well established as key regulators of learning and memory [34,97], as well as in the plasticity associated with substance use [35,36,98,99].

Post-transcriptional regulation also contributes to the expression of GABA_A_R subunits. The 3′UTRrome of the genes coding for *α* subunits is longer than the general 3′UTRome in mice, leading to the availability of many binding sites for RBPs and miRNAs to exert the translational control of expression [1]. A small number of RBPs have been identified to regulate the translation of GABA_A_R subunit mRNAs into protein, including Pum2 [63,177], NONO [64,65] and FMRP [55,67,68,69,70,71,72,73,74]. Multiple binding sites for miRNAs have been identified on the 3′UTR of GABA_A_R subunit genes. miRNAs are implicated in experience-dependent plasticity at the 3′UTR, and are emerging as a candidate in the development of substance use disorders by changing the expression of the genes involved in dependence and withdrawal [75,76,77,137,138].

BZs and alcohol act upon GABA_A_Rs, and both cause changes in the expression of subunits, as well as epigenetic modifications such as increased histone acetylation through the inhibition of HDACs [36,117,138,174]. The most consistent changes identified across both BZ and alcohol use in both human subjects and animal models include alterations in the expression of *α*1, 2, 4, and *γ*2 subunits [113,116,117,118,120,121,122,123,126,136,141,149,151,155,156]. While changes in the expression and regulation of multiple subunits has been identified in alcohol use disorder and animal models, the *α*2 subunit has also been genetically linked to alcoholism [108,149,151,152]. The *α*2 subunit is known to mediate the anxiolytic effects of alcohol, and is also upregulated in the amygdala during withdrawal states characterized by increased anxiety [118,123,174]. The use of other commonly abused substances such as stimulants also modifies the expression of GABA_A_R subunits, again with the *α*2 subunit identified by genetic, human postmortem and animal model studies [164,165,166,167,168,169,170,171,172]. While RBP binding sites have been identified for the *α*2 subunit, specific miRNAs and RBPs that may be affecting expression require further examination, particularly in the context of substance use. Identifying the factors upstream of expression regulation could possibly lead to better, more specific therapies for substance use disorder targeting the 3′UTR, as well as a further understanding of the factors regulating gene expression during acute and chronic exposure to drugs, and subsequent withdrawal.

## Figures and Tables

**Figure 1 ijms-21-04445-f001:**
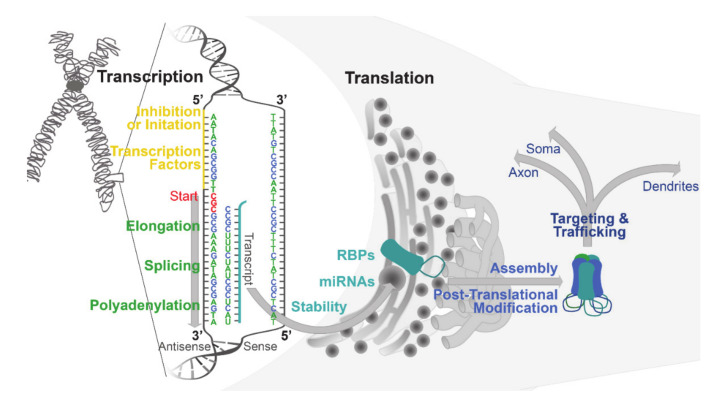
Levels of regulation impacting GABA_A_R expression and function. Expression of a GABA_A_R subunit begins with the initiation of transcription, which is controlled by a multiprotein complex that regulates the 5′UTR flanking genes encoding individual subunits. Transcription factors and other proteins that regulate transcription (yellow) bind to consensus sites in the 5′UTR and either promote or repress initiation. Many GABA_A_R subunit 5′UTRs contain consensus sites for more than one transcription factor, as well as multiple transcription start sites (red). Additional phases of splicing and polyadenylation (green) occur in the nucleus prior to the shuttling of the transcript to the rough endoplasmic reticulum for translation at ribosomes (grey circles). Translation of the subunit transcripts has been shown to be regulated by RNA-binding proteins (RBPs; turquoise), as well as microRNAs (miRNAs; turquoise). Once translated, the subunits are assembled (blue) together in quaternary structure to form a functional heteropentameric GABA_A_R. GABA_A_Rs are also subject to regulation via post-translational modifications (blue) such as phosphorylation and palmitoylation, as well as regulated targeting and trafficking to become preferentially localized to specific subcellular domains.

**Figure 2 ijms-21-04445-f002:**
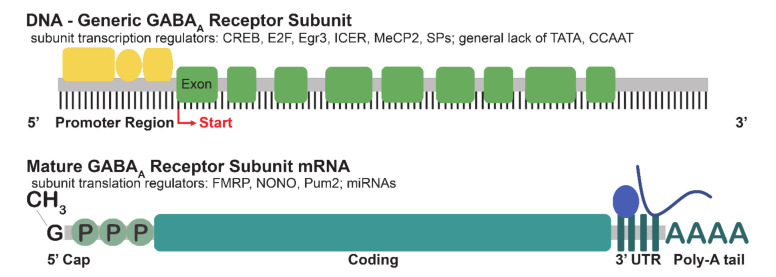
Overview of the major regulators of transcription and translation influencing the expression of GABA_A_R subunits. Generally, the 5′ UTR of GABA_A_R subunit genes contains multiple consensus sites for transcription factors (yellow), and in many cases, multiple transcription start sites (red). The majority of subunits lack canonical TATA and CCAAT sequences in their 5′UTRs. A number of regulators have been found to influence the transcription of multiple subunits including CREB and MeCP2. Maturation of the mRNA involves the addition of a 5′ cap as well as a poly-A tail adjacent to the 3′UTR. Mature GABA_A_R transcripts are transported out of the nucleus to ribosomes on the rough endoplasmic reticulum, where translation is influenced by RNA-binding proteins (blue oval) such as FMRP, NONO, and Pum2, as well as by microRNAs (miRNAs; blue line).

**Table 1 ijms-21-04445-t001:** Summary of the regulation of GABA_A_ receptor subunits with benzodiazepine (BZ) and alcohol use, and withdrawal.

	Benzodiazepine (BZ) Use	Alcohol Use
Transcription	↓ GABA_A_R *α*1 and *γ*2 subunit mRNA with chronic exposure [116] ↓ acetylated histone HC at *α*1 subunit promoter and ↑ HDAC 1 and 2 [117] ↑ *α*4, ↓ *α*1, ↓ *γ*2 mRNAs upon withdrawal [118]	↑ increase BDNF (known regulator of subunit transcription) [119] ↑ *α*1 mRNA in frontal cortex of human alcoholics [120,121] ↓ *α*1 and ↑ *α*4 mRNA chronic exposure [122] ↑ *α*2, *α*3, and *α*4 subunit mRNAs upon withdrawal [123]
Post-Translational Modification	↑ GABA_A_R uncoupling at BZ site [124,125]	↓ *β* subunit and *γ*2 subunit phosphorylation [120]
Total Expression	↑ *α*4 ↓ *α*1, and ↓ *γ*2 total protein upon withdrawal [118]	↑ *α*2 expression and ↓ *α*1 and ↓ *γ*2 with acute and chronic doses [126]
Surface Expression	↑ GABA_A_R internalization [113] ↓ GABA_A_R *α*2 surface expression with 24 h exposure [113]	↑ GABA_A_R internalization [120]

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
