# Peer review of "Regulation of GABAA Receptor Subunit Expression in Substance Use Disorders"

_ijms, 2020, doi:10.3390/ijms21124445_

Round 1

Reviewer 1 Report

In this article “Regulation of GABAA receptor subunit expression in substance use disorders” the authors review recent progresses on the role of GABAA receptor subunits expression in alcohol and benzodiazepine abuse. The topic is very interesting considering the role of GABAA system in the neurobiology of drug abuse and addiction. Unfortunately, in the actual form, this review miss the opportunity to provide important and useful informations for the readers. My main concerns are reported below:

  1. The first half of the paper is a detailed description of the molecular biology of the GABAA. Although interesting in general, this part lack a direct focus to the neurobiology of substance use disorders.
  2. In general many important and related papers are ignored and not included in this review. For example the papers that demonstrate the role of the alfa1 and delta subunits of GABAA in alcohol-motivated behaviours (see PMID 12626647, 12490572, 15451402, 16710315, 11781502). Other interesting articles that could have taken into consideration for the present review showed changes in GABAA subunits expression following ethanol chronic exposure in animal and clinical samples (PMID 15280440, 25278838). More papers are mentioned below.
  3. The title is too broad. Only benzodiazepine and alcohol abuse are discussed whereas GABAA receptor subtypes heterogeneity has been associated also to other drug of abuse such as nicotine (PMID 18482426, 29690521), cocaine (PMID 20133874, 22253714, 17382295) and amphetamine (PMID 26475507).
  4. This review do not describe the relationships among the various GABAA subunits expression in different brain nuclei and the significance for drug of abuse. For example GABAA receptor alpha4 subunit in the nucleus accumbens shell mediates alcohol intake (PMID 19144854), ethanol self-administration led to significant changes in GABAA subunits mRNA expression in different cortical areas (PMID 17117962) and chronic cocaine alter the gene expression of alfa2 subunits in the nucleus accumbens (PMID 17382295).
  5. Finally, the findings about the GABAA receptor subtypes described in the paper are never associated to the many significant facets of drug abuse and addiction such as reward, self-administration, drug motivated behaviours, anxiety and stress. Overall this makes difficult to understand the neurobiological and clinical significance of the findings.

Author Response

1. The first half of the paper is a detailed description of the molecular biology of the GABAA. Although interesting in general, this part lack a direct focus to the neurobiology of substance use disorders.

We have shortened this portion of the manuscript by removing superfluous details, as well as acronyms / jargon. We have also now added several sentences throughout this section that link the basal mechanisms regulating GABAAR subunit expression to substance use disorders, helping to regain the focus on substance use.

2. In general many important and related papers are ignored and not included in this review. For example the papers that demonstrate the role of the alfa1 and delta subunits of GABAA in alcohol-motivated behaviours (see PMID 12626647, 12490572, 15451402, 16710315, 11781502). Other interesting articles that could have taken into consideration for the present review showed changes in GABAA subunits expression following ethanol chronic exposure in animal and clinical samples (PMID 15280440, 25278838). More papers are mentioned below.

We have now read and synthesized the results of these papers to include them in the review, and appreciate the reviewer bringing them to our attention.

3. The title is too broad. Only benzodiazepine and alcohol abuse are discussed whereas GABAA receptor subtypes heterogeneity has been associated also to other drug of abuse such as nicotine (PMID 18482426, 29690521), cocaine (PMID 20133874, 22253714, 17382295) and amphetamine (PMID 26475507).

We have now added a short section on stimulants to include the modifications in GABAAR expression that are induced by this highly abused class of substances. We also have added some information on opiates. These additions make the review broader and more consistent with the title, and also add important information to the topic. Again, appreciative of these suggestions.

4. This review do not describe the relationships among the various GABAA subunits expression in different brain nuclei and the significance for drug of abuse. For example GABAA receptor alpha4 subunit in the nucleus accumbens shell mediates alcohol intake (PMID 19144854), ethanol self-administration led to significant changes in GABAA subunits mRNA expression in different cortical areas (PMID 17117962) and chronic cocaine alter the gene expression of alfa2 subunits in the nucleus accumbens (PMID 17382295).

We have now added details to specify brain region more clearly throughout, and also added text that indicates that some of the changes are region specific. We also indicate that attention to region and the relationship between brain region and specific aspects of addiction and withdrawal are important for further study.

5. Finally, the findings about the GABAA receptor subtypes described in the paper are never associated to the many significant facets of drug abuse and addiction such as reward, self-administration, drug motivated behaviours, anxiety and stress. Overall this makes difficult to understand the neurobiological and clinical significance of the findings.

Certainly, the ability to specifically attribute changes in GABAAR subunit expression with these facets of substance use would be a very powerful advance. In order to address this, we have added text to our concluding statements that this would be a desirable goal, and make other suggestions about how we can continue to advance this area of work. In reexamining the literature, we were unable to find sufficient resolution in many of the studies already explored to delineate these facets. Due to the deep focus on GABAARs we are also unable to add the significant amounts of text needed to fully explore these important functional topics in the present review.

Reviewer 2 Report

The authors undertook to summarize / review a very important and very diverse field in the manuscript. Of course, for any review, it is difficult to find a balance between the essential disclosure of background information and compliance with space limits. In this manuscript, this is solved in the „Substance Use” chapter better, less so in the introductory part. The introduction is difficult to interpret in certain sections; it is difficult to follow the line of thought of the text due to many abbreviations and function markings. Of course, it is essential to provide these, but it would be good to find some compromise. It might be possible to include a table that, in addition to the abbreviations, indicates the functions, which could be removed from the text. This is especially true for the Transcription and Translation chapters. It is important to emphasize that abbreviations are not used uniformly, in some places the abbreviation is given at the first mention, in others the explanation is missing or only given much later (for example, Egr3 is mentioned in row 149 but becomes clear in row 284). It would be worth rethinking the structure and focus of the manuscript: in case of such a complex topic, it is particularly difficult to decide what is worth explaining in more detail and what in short. This was better done in the „Substance Use” chapter of this manuscript, but not really in the introduction sections. If the size limit allows, some concise explanatory sentences should be inserted to make the topic clear. Otherwise, fewer topics will need to be presented. The references are correct; a more precise definition of substance use disorder would be advantageous. There are very few typing errors in the manuscript (like chromosome X in row 142).

Author Response

The authors undertook to summarize / review a very important and very diverse field in the manuscript. Of course, for any review, it is difficult to find a balance between the essential disclosure of background information and compliance with space limits. In this manuscript, this is solved in the „Substance Use” chapter better, less so in the introductory part. The introduction is difficult to interpret in certain sections; it is difficult to follow the line of thought of the text due to many abbreviations and function markings. Of course, it is essential to provide these, but it would be good to find some compromise. It might be possible to include a table that, in addition to the abbreviations, indicates the functions, which could be removed from the text. This is especially true for the Transcription and Translation chapters. It is important to emphasize that abbreviations are not used uniformly, in some places the abbreviation is given at the first mention, in others the explanation is missing or only given much later (for example, Egr3 is mentioned in row 149 but becomes clear in row 284).

We have now shortened the introductory portion of the manuscript, by removing extra details that are not necessary, as well as by removing abbreviations and jargon that detract from the readability. We have also checked our abbreviation use and corrected some errors including that noted for Egr3.

It would be worth rethinking the structure and focus of the manuscript: in case of such a complex topic, it is particularly difficult to decide what is worth explaining in more detail and what in short. This was better done in the „Substance Use” chapter of this manuscript, but not really in the introduction sections. If the size limit allows, some concise explanatory sentences should be inserted to make the topic clear. Otherwise, fewer topics will need to be presented.

We have now restructured the flow of the manuscript, nesting some sub topics within more broad headings. We have also removed extra details, particularly from the first half of the manuscript. In addition, we have also added several explanatory sentences to the first half of the manuscript that link the basal mechanisms regulating GABAAR subunit expression to substance use disorders, helping to regain the focus on substance use.

The references are correct; a more precise definition of substance use disorder would be advantageous.

We have now added a sentence that defines substance use disorder before the DSM description in the section that introduces substance use disorders.

There are very few typing errors in the manuscript (like chromosome X in row 142).

We have changed “the x chromosome” to “chromosome X”, we also corrected some other typing errors that we found upon an additional thorough examination.

Reviewer 3 Report

This manuscript focuses attention on a topic of great interest both from a research point of view to understand the mechanisms regulating GABAA receptor expression including transcription and translation, and from a medical point of view regarding substance use disorders. Regulation of GABAA receptor subunit expression is critical in the modulation of strength of synaptic inhibition. Changes in the GABAA receptor subunit expression and its dysregulation critically affect brain functions. The study of the regulation of GABAA receptor subunit expression in substance use disorders could be helpful in the development of novel targeted therapies. This manuscript is well structured and solid, and will be of value to the field.

Among various substance use disorders, this manuscript describes the benozodiazepine and alcohol use. What about effects of other substances such as opioids and cannabis on GABAA receptor subunit expression?

Author Response

This manuscript focuses attention on a topic of great interest both from a research point of view to understand the mechanisms regulating GABAA receptor expression including transcription and translation, and from a medical point of view regarding substance use disorders. Regulation of GABAA receptor subunit expression is critical in the modulation of strength of synaptic inhibition. Changes in the GABAA receptor subunit expression and its dysregulation critically affect brain functions. The study of the regulation of GABAA receptor subunit expression in substance use disorders could be helpful in the development of novel targeted therapies. This manuscript is well structured and solid, and will be of value to the field.

Among various substance use disorders, this manuscript describes the benozodiazepine and alcohol use. What about effects of other substances such as opioids and cannabis on GABAA receptor subunit expression?

We have now added a short section on other highly abused substances which includes stimulants and opiates. We were not able to locate studies that characterized subunit expression following cannabis exposure.

Round 2

Reviewer 1 Report

The article has substantially improved. I think that it is now of appreciable interest for the readers.